# A Mitochondrial Perspective on Noncommunicable Diseases

**DOI:** 10.3390/biomedicines11030647

**Published:** 2023-02-21

**Authors:** Yifan Zheng, Jing Zhang, Xiaohong Zhu, Yuanjuan Wei, Wuli Zhao, Shuyi Si, Yan Li

**Affiliations:** 1Key Laboratory of Antimicrobial Agents, Institute of Medicinal Biotechnology, Chinese Academy of Medical Sciences and Peking Union Medical College, Beijing 100050, China; 2NHC Key Laboratory of Antibiotic Bioengineering, Laboratory of Oncology, Institute of Medicinal Biotechnology, Chinese Academy of Medical Sciences and Peking Union Medical College, Beijing 100050, China

**Keywords:** mitochondria, mitochondrial dysfunction, OXPHOS, mitochondrial diseases

## Abstract

Mitochondria are the center of energy metabolism in eukaryotic cells and play a central role in the metabolism of living organisms. Mitochondrial diseases characterized by defects in oxidative phosphorylation are the most common congenital diseases. Meanwhile, mitochondrial dysfunction caused by secondary factors such as non-inherited genetic mutations can affect normal physiological functions of human cells, induce apoptosis, and lead to the development of various diseases. This paper reviewed several major factors and mechanisms that contribute to mitochondrial dysfunction and discussed the development of diseases closely related to mitochondrial dysfunction and drug treatment strategies discovered in recent years.

## 1. Introduction

In eukaryotes, mitochondria, organelles necessary for the maintenance of normal cellular physiological functions and production of adenosine triphosphate (ATP) via tricarboxylic acid cycle and fatty acid oxidation for cellular energy supply, function as the primary site of energy metabolism. The intermediate metabolites, during energy production, support an organism’s various physiological functions, being inextricably linked to processes such as calcium ion dynamic balance, redox dynamic balance, apoptosis regulation, and amino acid and iron/sulfur cluster synthesis [1] (Figure 1).

As cellular stress sensors, the number and function of mitochondria maintain homeostasis in the face of environmental stress at the organelle and molecular levels, helping cells adapt to their environment. Mitochondrial dysfunction, such as the rupture of the mitochondrial complex, mitochondrial uncoupling, and cristae remodeling and swelling, leads to an increase in Reactive oxygen species (ROS), energy stress, and cell death [2]. ROS plays a physiological role in intracellular signaling pathways when kept at a low level [3]. Mitochondria are the primary site of ROS production, and individual electrons that escape during ATP production through Oxidative Phosphorylation (OXPHOS) cause the production of superoxide anion radicals, a by-product [4]. When the antioxidant capacity of mitochondrial antioxidant enzymes (glutathione peroxidase, manganese superoxide dismutase, etc.) is reduced or ROS production is increased, oxidative stress, mitochondrial respiratory enzyme activity inhibition, mtDNA, protein, and lipid damage, and mitochondrial dysfunction occur [5,6]. As semiautonomous organelles, the function and biosynthetic processes of mitochondria are regulated jointly by nuclear and mtDNA. Compared with the nuclear gene, mtDNA, which is located in the mitochondrial IMM, is more susceptible to ROS attack and lacks an effective damage repair system. Mutations of mtDNA at any locus may affect important functional regions of the genome, leading to tissues and organs developing abnormalities and clinical phenotypes.

Many human diseases are related to mitochondrial dysfunction. Both mutation of the nuclear gene and mtDNA may lead to mitochondrial disorders that can be passed on to offspring through Mendel’s law inheritance for nDNA mutations and maternal inheritance for mtDNA, and these diseases are classified as primary mitochondrial diseases. Those of nongenetic origin are referred to as secondary mitochondrial diseases, and among them, noncommunicable chronic diseases (NCDs) get more attention than noninfectious ones because they collectively pose a huge challenge for public health systems in the world [7]. An in-depth understanding of the relationship between mitochondrial dysfunction and these diseases will help in the treatment of related diseases. This review focuses on the role of mitochondrial dysfunction in the occurrence and development of NCDs, as well as mitochondrial-targeted drugs or treatment strategies for these diseases.

## 2. Mitochondrial Dysfunction and NCDs

Different tissues and organs in the body have different energy requirements and differ in their susceptibility to mitochondrial dysfunction; that is, skeletal muscle, brain, heart, and other organs with high energy requirements are more sensitive. A combination of genetic and environmental factors, such as low-grade inflammation and oxidative stress, contributes to the development of NCDs, and many of these factors have in common the ability to either damage mitochondria or interfere with mitochondrial repair [7]. Mitochondria dysfunction plays an important role in the pathophysiology of many NCDs (Table 1).

### 2.1. Neurodegenerative Diseases

Normal metabolism of neuronal cells requires large amounts of energy from mitochondria, which are involved in cellular energy supply, generation of ATP, regulation of ROS concentration and calcium homeostasis, as well as in action potential generation and excitatory neuron transmission [8]. Mitochondrial dysfunction in neuronal cells often leads to pathologies in the body.

Parkinson’s disease (PD), a degenerative disease of the central nervous system, is characterized by degenerative cell death of dopaminergic (DA) neurons in the nigrostriatal region of the midbrain. An early sign of PD progression is mitochondrial dysfunction, a key link in the pathogenesis of PD. Kann et al. found that healthy cells transfected with mtDNA isolated from focal areas in the brain of patients with PD exhibited biochemical features of PD. Animal models exhibited mitochondrial complex I deficiency and other neuropathological features of PD when injected with mitochondrial toxins MPTP and rotenone [9]. In patients with PD, mtDNA deletion variants in the basal ganglia are significantly high [10]. Mutations in genes coding proteins linked to PD, such as DJ-1, parkin, PINK1, alpha-synuclein, and LRRK2, affect mitochondrial function and integrity, causing enhanced ROS generation and vulnerability to oxidative stress [11,12,13,14]. TMEM175 motif variants in mtDNA impair lysosomal and mitochondrial function and promote α-synuclein deposition [15]. In addition, mutations can cause transcriptional inhibition of Parkin-interacting substrate PGC-1α and impair mitochondrial biogenesis, which cumulatively results in the loss of DA neurons [16]. In animal models of PD, PGC1α knockout mice exhibited stronger neurodegeneration and motor abnormalities in striatal neurons [17]. In cellular disease models, PGC1α protein overexpression led to increased expression of nuclear coding subunits of the mitochondrial RC, blocking the loss of DA neurons. PGC- 1α can be a potential therapeutic target for early intervention in PD. Currently, the role of antioxidant neurotrophic strategies in PD treatment is also emphasized.

Alzheimer’s disease, a neurodegenerative disorder, is characterized by progressive cognitive decline culminating in dementia, with extensive deposition of β-amyloid plaques and subsequent tau protein aggregation in the neocortex, mediating neuroinflammation and leading to neurodegeneration. It is still not clear whether mitochondrial dysfunction plays a direct role in the initiation of AD, but oxidative damage and neuroinflammation have been shown to correlate with AD progression. The pathology of Alzheimer’s disease can be visualized in transgenic mice that express Aβ plaques and neurogenic fiber tangles, exhibiting significant mitochondrial abnormalities such as reduced complex I and IV activity, decreased mitochondrial membrane potential, and increased free radical production. Impaired mitochondrial membrane function also affects intracellular calcium homeostasis, which may be related to the abnormal processing of neuronal calcium found in patients with AD and AD models [18]. In addition, multiple proteins in mitochondria are closely related to the pathological process of AD. Hyperphosphorylation of the neurotoxic tau proteins produce aggregates deposited in the neocortex, inducing synaptic loss and hampering the function of neural networks, which is key to the development of clinical AD [11]. Phosphatidylethanolamine synthesis mediated by mitochondria-associated membranes in lipid metabolism regulates tau phosphorylation [19]. Acyl coenzyme A cholesterol acyltransferase, essential for the β-amyloid formation, is extensively present in mitochondrial membranes [20]. The OMM protein TOMM40, which regulates the entry of cytoplasmic proteins into mitochondria, is closely associated with AD pathogenesis [21]. Many therapies targeting mitochondrial dysfunction in neurodegeneration and cognitive dysfunction in AD rely on the application of antioxidants and a reduction in free radical levels [22,23].

Amyotrophic lateral sclerosis is a progressive neurodegenerative disease affecting motor neurons in the brain and spinal cord. The role of mitochondrial dysfunction in the pathogenesis of ALS is unclear. However, the mutation in the cytoplasmic Superoxide Dismutase (SOD1), located in the mitochondria, may be related to some familiar forms of ALS [18]. Expressing mutant SOD1 targeted to mitochondria is sufficient to produce loss of motor neurons and an ALS phenotype [18,21].

### 2.2. Cardiovascular Diseases(CVDs)

The heart is a highly energy-consuming organ; therefore, cardiomyocytes contain abundant and active mitochondria. Mitochondrial dysfunction involves energy deficiency, increased production of OXPHOS-related harmful catabolic products, increased autophagy and apoptosis, and metabolic abnormalities that lead to the development CVDs.

In CVDs, mitochondria are damaged due to membrane rupture and matrix depletion, resulting in decreased activity of complex I and IV and damaged respiratory chain, leading to insufficient ability to synthesize ATP [24,25]. Low-density lipoprotein(OxLDL), one substance closely related to the occurrence and development of CVDs, leads to mitochondrial damage with increased complex I activity and oxidative stress [26], which in turn leads to excessive ROS production and calcium dysregulation [27]. Excessive calcium ion concentration in mitochondria may result in the opening of mitochondrial permeability transition pores (MPTPs) present on the inner layer of the mitochondrial membranes; this phenomenon causes cellular apoptosis and is an important factor in the occurrence of ischemia-reperfusion myocardial injury [28]. In the context of ischemia-reperfusion injury, cellular bioenergy is deficient, and serine/threonine kinase casein kinase-2α promotes phosphorylation of mitochondrial fission factor, the receptor for dynamin-related protein 1(Drp1), leading to mitochondrial division and cell death [29]. In the hearts of patients with nonischemic coronary microvascular disease, ROS concentrations are also twice as high as in healthy hearts [30]. Pathological remodeling of cardiomyocytes occurs, resulting in altered cardiac metabolism with increased glycolysis and decreased fatty acid oxidation [31]. This outcome, in turn, leads to inadequate energy supply and cardiac dysfunction.

At present, it has been found that many regulatory factors closely related to the occurrence and development of CVDs are closely related to the generation and function regulation of mitochondria in cardiac myocytes. PGC1α (PPARγ coactivator-1 α) is a transcriptional coactivator that mediates many biological programs related to energy metabolism, which is involved in the development and progression of CVDs such as cardiac hypertrophy and diabetic cardiovascular complications by regulating mitochondrial function in cardiomyocytes. PGC-1α promotes mitochondrial biogenesis by activating downstream nuclear respiratory factors NFR-1 and NRF-2, mitochondrial transcription factor TFAM, and mtDNA-encoded oxidative phosphorylated protein transcription. Mitochondrial biogenesis is critical for the regulation of mitochondrial turnover and function, and the complex interaction between mitochondrial function and metabolism and epigenetic signaling perturbations leads to a considerably increased risk of phenotypic disorders in the vasculature and, thus, CVD [32]. Dysregulation of PGC-1α signaling during heart failure occurs at the transcriptional and post-transcriptional levels, contributing to the progression of cardiac dysfunction through multiple mechanisms [33]. Recently, protein homeostasis has emerged as a potential mechanism for maintaining cardiac mitochondrial stability. Protein homeostasis controls the biogenesis, folding, and degradation of mitochondrial proteins; this process is disrupted during stress in cardiovascular system cells [34]. First, when the number of misfolded mitochondrial proteins increases, the activation of transcription factor 5 (ATF5) enhances the activation of mtUPR. The ATF5 then translocates to the nucleus and stimulates the upregulation of genes that fold mitochondrial proteins and restore protein stability; stimulation of mtUPR improves mitochondrial function and reduces cardiac damage caused by myocardial ischemia-reperfusion injury [35]. Second, changes in specific enzymes and proteins that are highly correlated with mitochondrial dynamics are commonly observed in various CVDs, such as ischemic heart disease and heart failure. For example, mitochondrial fusion protein 1(mitofusin, Mfn1) phosphorylation which promotes mitochondrial fusion, and Mfn2 ubiquitination which leads to mitochondrial fragmentation, can be observed in cells with myocardial ischemia-reperfusion injury [36].

### 2.3. Tumors

As an important organ for energy metabolism, cell stress regulation, and cell function regulation, such as energy metabolism, calcium homeostasis, redox regulation, and apoptosis, the relationship between mitochondria and cancer and mitochondria-targeted therapy of cancer has been a hot research topic in recent decades. A large number of studies have confirmed the role of the mitochondrion in the process of tumor development, invasion, and malignancy.

Warburg effect, which is well known in that cancer cells in culture catabolize glucose by lactic fermentation even in the presence of oxygen, is the major metabolic hallmark of cancer, suggesting a potential role for mitochondrial metabolism in cancer [37]. However, the relationship between aerobic glycolysis and mitochondrial damage has been controversial. In different tumor cells, oxidative phosphorylation damage of mitochondria is also different, and most cancers still retain mitochondrial respiration and other functions [38,39,40]. However, at the same time, it is also proposed in the literature that inhibition of oxidative phosphorylation (OXPHOS) leads to elevated glycolytic metabolism [41]. Reversing the Warburg effect might be an approach to restore cell differentiation in cancer. Haowen Jiang et al. used a mitochondrial uncoupler, niclosamide ethanolamine (NEN), to activate mitochondrial respiration, which induced neural differentiation in neuroblastoma cells [42]. The results suggest that mitochondrial uncoupling is an effective metabolic and epigenetic therapy for reversing the Warburg effect and inducing differentiation in neuroblastoma. (Mitochondrial uncoupling induces epigenome remodeling and promotes differentiation in neuroblastoma.). On the other hand, Weier Bao et al. used a core-shell type of nanoagent with iron (III) carboxylate metal-organic frameworks (MOFs) as shell while upconversion nanoparticles (UCNPs) as core, which enables near-infrared (NIR) light-triggered synergistically reinforced oxidative stress and calcium overload to mitochondria [43]. This agent destroys the mitochondria in tumor cells and then effectively kills tumor cells and inhibits tumor growth. The dual-mitochondria-damage-based therapeutic potency of the nanoagent has been unequivocally confirmed in cell- and patient-derived tumor xenograft models in vivo. (MOFs-Based Nanoagent Enables Dual Mitochondrial Damage in Synergistic Antitumor Therapy via Oxidative Stress and Calcium Overload) Therefore, the impact of mitochondrial oxidative damage on the occurrence and development of different tumors and targeted antitumor strategies still need more research.

Mitochondrial dynamics also contribute to the occurrence and development of tumors which excessive division and reduced fusion are the characteristics of many tumors [44,45,46]. In some tumor cells, the increased expression and phosphorylation of mitochondrial mitosis-related protein Drp1 and the decreased expression of mitochondrial fusionprotein MFN2 can be observed, but the trigger factor and mechanism are still unclear [47,48]. Mdivi1 and Drp1 inhibitors can inhibit tumor cell proliferation and induce cell apoptosis [49].

The rapid proliferation of tumor cells requires accelerated metabolism, producing large amounts of amino acids, nucleotides, and lipids through aerobic glycolysis and glutamine catabolism [50]; through the TCA cycle, mitochondria play a central role in glutamine catabolism. Several enzymes of the TCA cycle are often mutated or deregulated in human cancers, including aconitase, isocitrate dehydrogenase, fumarate hydratase, succinate dehydrogenase, and KGDHC [51,52]. Sirtuins are linked to glutamine metabolism and play both promoter and suppressor roles in tumor development. SIRT4, which is found in mitochondria, is a tumor suppressor that works by inhibiting glutamine metabolism and promoting genomic stability [53]. Silencing of SIRT3 and SIRT5 genes upregulate ROS levels, which in turn, leads to oxidative damage of cellular structures and activates apoptosis in cancerous tissues, making them significantly more sensitive to chemotherapy and radiotherapy [54,55,56]. Mitochondria regulate apoptosis primarily by altering the permeability of the mitochondrial membrane and mediating the release of apoptotic factors. In tumor cells, Bcl-2 is highly expressed, and the Bcl-2 family proteins interact with some proteins and regulate the permeability of the OMM [57]; this phenomenon, in turn, inhibits apoptosis caused by the aggregation of apoptotic precursor proteins and contributes to the sustained growth of tumor cells. Under environmental stress, the regulatory factor PGC-1α promotes tumor cell survival and metastasis by mediating mitochondrial biogenesis and OXPHOS. The PINK1–Parkin pathway, Bcl-2 protein-like 13, and others trigger mitosis drug resistance to increase cancer cell resistance to various commonly used chemotherapeutic agents such as 5-fluorouracil (5-Fu), cisplatin, and adriamycin (Dox) [58,59].

In the study of mitochondrial-targeted anti-tumor strategies, in addition to the discovery of drugs targeting the key regulatory factors of mitochondrial function, there are also many studies on the role of mitochondrial metastasis in tumor treatment. Chao Sun et al. reported that transferring normal human astrocytic mitochondria into glioma cells rescues aerobic respiration and enhances radiosensitivity [60]. Jui-Chih Chang et al. found that mitochondria transplantations in MCF-7 cells induced cell apoptosis, inhibited cell growth, and increased the cellular susceptibility of both the MCF-7 and MDA-MB-231 cell lines to Doxorubicin and Paclitaxel [61].

### 2.4. Diabetes

Mitochondrial dysfunction has been found to be an important factor affecting β-cell sensitivity to insulin, and impaired mitochondrial oxidation, reduced biogenesis, and excessive ROS production are common features of diabetes [62]. Mitochondria aid in glucose signaling to release insulin, and persistent hyperglycemia increases the metabolic demands of pancreatic β-cells, leading to excessive and sustained ROS production; this result, in turn, increases lipid peroxide and isoprostanes production and leads to DNA damage and ultimately to β-cell apoptosis [35]. The Krebs cycle contributes to the overproduction of electron donors such as NADH and flavin adenine dinucleotides; these donors exacerbate ROS production, leading to an increased proton gradient in the IMM, resulting in oxidative damage in pancreatic β-cells from patients with type 2 diabetes [35,54]. Various transcription factors such as PGC1α, peroxisome proliferator-activated receptor (PPAR), and nuclear respiratory factor (NRF) regulate mitochondrial biogenesis. In pancreatic β-cells, mGPDH and glycerophosphate shuttle mediate the glucose-dependent insulin secretion process, and the accumulation of damaged or depolarized mitochondria in β-cells is associated with oxidative stress and the development of diabetes mellitus. Mitochondrial dysfunction can lead to a compensatory increase in fatty acid oxidation in adipocytes, resulting in increased levels of acetyl coenzyme a and NADH; this, in turn, induces adipocyte apoptosis [36], triggers M1 macrophage proliferation and inflammatory mediator release and exacerbates insulin resistance [63]. Excess free fatty acid (FFA) accumulation in adipocytes activates NADPH oxidase and increases ROS production. Oxidative stress caused by mitochondrial dysfunction in adipocytes and skeletal muscle cells plays an important role in the pathogenesis of insulin resistance and type 2 diabetes [50,64]. Excessive accumulation of FFA in myocytes increases the risk of insulin resistance [65,66]. Mitochondria are dynamic organelles, and the maintenance of a balance between fusion and fission is important to maintain the normal biological function of mitochondria. Mitochondrial fusion is mediated by Mfn1, Mfn2, and optic atrophy protein 1(Opa1), while fission is controlled by proteins such as mitochondrial fission factor (MFF) and Drp1 [67,68]. Fluctuations in the levels of proteins related to mitochondrial dynamics have been widely observed in diabetic patients. Reduced Mfn2 expression has been observed in type 2 diabetic patients in close association with skeletal muscle mitochondrial dysfunction. In addition, phosphorylation of the Drp1 serine 616 position and activation of Drp1, upregulation of Mfn1 and Mfn2 expression, and improvement of both fat oxidation and insulin sensitivity in vivo were observed in insulin-resistant patients who performed moderate amounts of aerobic exercise [53]. Therefore, drug development targeting interventions such as mitochondrial biogenesis, fusion, and fission kinetics would be an effective way to treat diabetes.

**Table 1 biomedicines-11-00647-t001:** Mitochondrial dysfunction in common diseases and possible mechanisms.

Disease	Mitochondrial Disorder	Related Tissues or Cells	Mechanisms that May Cause Those Disorders	References
Parkinson’s disease	Mitochondrial complex I deficiency; Higher mtDNA deletion variants; Defective mitochondrial autophagy.	Basal ganglia; DA neurons	Mfn1 ubiquitination and S-nitrosylation of DRP1; α-synuclein aggregation; Mutations in PINK1 and Parkin.	[9,14,23,65]
Alzheimer’s disease	Com-plex I and IV activity decline; Decreased mitochondrial membrane potential; Increased free radical production	Neocortex	Impaired mitochondrial membrane function; Aβ plaques deposition.	[18,20]
Cardiovascular diseases	Mitochondrial ATP synthesis capacity reducing. Complex I activity and oxidative stress increasing; Mitochondrial matrix calcium overload; Mitochondrial division;	Cardiomyocyte; Endothelial cells.	Mitochondrial membrane rupture and matrix depletion; Calcium overload; Phosphorylation of mitochondrial fission factor.	[24,26,29]
Tumors	Decrease of SIRT4 expression level; High levels of ROS production.	Malignant tumor cells; Undifferentiated tumor cells.	mGPDH abundance and activity elevated.	[53,66]
Diabetes	Excessive and sustained ROS production. The levels of acetyl coenzyme A and NADH increasing; Expression of Mfn2 reducing.	β-cell; Adipocyte; Skeletal muscle.	mtDNA mutations; Persistent hyperglycemia; Accumulation of depolarized mitochondria.	[69,70,71]

## 3. Therapeutic Strategies Using Mitochondria as Therapeutic Targets or Targets of Intervention

### 3.1. Oxidative Damage

Mitochondria are the primary site of ROS production and the first target of oxidative stress by ROS attack. Oxidative damage caused by mitochondrial oxidative stress is an important factor in causing mitochondrial dysfunction, which in turn induces cellular abnormalities and various diseases. Excessive mitochondrial ROS production can directly damage mitochondrial components (such as mtDNA, mitochondrial membrane, and respiratory chain proteins) and induce mitochondrial oxidative stress, impairing mitochondrial function [72]. Directly targeting mitochondrial oxidative stress to counteract mitochondrial oxidative damage is a very effective therapeutic strategy, and development directions include both directly counteracting the negative effects produced by excess ROS and enhancing mitochondrial antioxidant enzyme activity.

Endogenous ROS are mainly derived from many biochemical processes such as respiratory chain, OXPHOS, and TCA, and metabolic byproducts of enzymatic processes such as glycerol-3-phosphate dehydrogenase, monoamine oxidase, and cytochrome P450, while exogenous ROS are mainly caused by aging, inflammation, cancer lesions, and stress from irritating compounds such as alcohol and pesticides [73]. Clinical compounds with ROS scavenging properties, such as thiols, vitamins, Nrf2 activators, coenzyme Q10, and ubiquinone, have been developed. For example, glutathione (GSH), a very important thiol endogenous antioxidant in neuronal cells, can directly inactivate ROS, and microinjection of the GSH precursor N-acetyl-cysteine in the brain of rats can effectively increase their brain GSH levels and promote cognitive levels in rats by ameliorating excessive mitochondrial oxidation in neuronal cells [74]. Vitamin E achieves ROS inhibition by scavenging free radicals and inhibiting lipid peroxidation [75], and vitamin C inhibits mitochondrial oxidative stress in heavy metal-induced lipid peroxidation and ROS accumulation in neuronal cells [76]. In a double-blind trial of the treatment of endometriosis caused by oxidative stress through antioxidant vitamin supplementation, patients taking vitamin C and vitamin E had significantly lower levels of ROS and significantly less painful disease [77]. Currently, some common compounds that protect cells from oxidative stress in laboratory and clinical applications also include CoQ10, MitoQ, and so on (Table 2). Recent studies have revealed that glutathionylation modification of NDUFS1, a subunit essential for the activity of complex I, forms a feedback mechanism with glycerol-3-phosphate dehydrogenase-mediated ROS production [78] and targets reduction of the glutathionylated portion of NDUFS1 in complex I would be a promising strategy to reduce ROS production by glycerol-3-phosphate dehydrogenase and proline dehydrogenase. The use of compounds to stimulate massive ROS production inside tumor cells when treating tumors is also one of the common clinical strategies. Ion chelators can inhibit mitochondrial metabolism and respiratory rate to induce ROS production [79], such as the iron complexing agent VLX600, which inhibits mitochondrial OXPHOS and provides excellent resistance to ovarian, breast, and colorectal cancers [80]. Endogenous antioxidant enzymatic defense systems exist in aerobic organisms, and antioxidant enzymes, mainly including superoxide dismutase (SOD), Prdx3, catalase (CAT), and glutathione peroxidase (GPX), exert antioxidant effects. Epalifuzin is a drug with anti-inflammatory, antioxidant, anti-atherosclerotic, and anti-arrhythmic properties [81], which significantly increase the mRNA expression of antioxidant enzymes SOD1 and GPX1 and reduce the production of mitochondrial ROS [82]. Combination therapies are now commonly used clinically to counteract mitochondrial dysfunction and usually include vitamin B, which usually ameliorates complex I and complex III abnormalities, creatine that provides high-energy phosphates, and the antioxidants vitamin C, vitamin E, coenzyme Q10, and ubiquinone for delaying the effects caused by mitochondrial damage [83].

Although antioxidants, in the traditional sense, have a significant protective effect on damaged cells caused by mitochondrial dysfunction, most of them are not sufficiently bioavailable and are more difficult to cross the blood-brain barrier to achieve neuroprotective effects. In recent years, the use of nanomaterials with good biocompatibility and safety as mediators for targeted drug delivery has become a hot topic of research. Mitochondria-targeting nanocomposites gold-selenium core-shell nanostructures (AS-I/S NCs) modified with the antioxidant peptide SS31 can achieve mitochondria-targeted ROS inhibition [84]. The multifunctional heterogeneous peptide HNSS consisting of antioxidant peptide SS31 and neuroprotective peptide S14G-Humanin piggybacked on PEG-PTMC (Cit) nanoparticles enriched in mitochondria can effectively improve mitochondrial dysfunction by alleviating oxidative stress and restoring mitochondrial ultrastructure [85].

In the field of catalytic and chemical reaction nanomedicine, the use of metal element valence changes can effectively alter the level of ROS production for therapeutic purposes such as antitumors. Multivalent metal ions (e.g., Fe2+/Fe3+, Cu+/Cu2+, and Mo4+/Mo5+/Mo6+) are the basis for ROS explosive production [86,87,88], and multivalent metal-based nanomaterials (nanoenzymes) with peroxidase-like activity can promote harmful ROS production, induce mitochondrial dysfunction and promote cell death for tumor therapy. Ling et al. designed MoO3-xNUs to target the tumor environment to induce CAT-like reactions and generate cytotoxic superoxide radicals to induce apoptosis in tumor cells [89]. Coated organic backbone AHT-Ce/SrMOF containing Ce and Sr metal ions implanted in rats exhibited superoxide dismutase and catalytic peroxidase-like activity, activated AMPK signaling pathway, catabolized ROS in bone-related mesenchymal stem cells (BMSCs) and restored their mitochondrial function, and reversing the cellular senescence process [90]. At the same time, CaF2, a metal compound with peroxidase-like activity, can also be used as a nanoenzyme for tumor therapy. Dong et al. used harmless ultrasound as an external energy source to break through the tissue penetration depth and amplify the peroxidase-like catalytic activity of CaF2 to promote harmful ROS production and induce mitochondrial dysfunction induced by calcium overload in tumor cells and exerted significant antitumor effects in mammary tumors and hepatocellular carcinoma. The significant antitumor effects in animal models have broadened the ideas for nanoenzyme research [91].

**Table 2 biomedicines-11-00647-t002:** Antioxidants targeting mitochondrial oxidative damage.

Type	Name	Mechanism of Action	Therapy Area	Highest R&D Status (Global)	References
ROS scavenger/ROS generation inhibitor	MitoQ	Blocks the generation of ROS and mitochondrial protein thiol oxidation	Nervous system diseases; Endocrinology and metabolic disease; Tumors; Cardiovascular diseases	Clinical Phase II	[92,93,94]
	SKQ-1	Electron transport chain complex proteins regulator	Nervous system diseases; Endocrinology and metabolic disease	Listed Drug	[95,96]
	SS31	Restriction of mitochondrial ROS production	Nervous system diseases; Tumors; Cardiovascular diseases	New drug application	[97,98,99]
	GSH	Directing inactivation of ROS	Nervous system diseases; Tumors; Cardiovascular diseases	Listed Drug	[74,100,101]
	Ascorbic acid	Quenching mitochondrial ROS; Mimicing the functions of mitochondrial SODs	Cardiovascular diseases; Endocrinology and metabolic disease	Listed Drug	[102,103]
	α-tocopherol	Scavenging of lipid peroxyl radicals	Nervous system diseases; Tumors	Listed Drug	[104,105]
	Coenzyme Q10	Scavenging of ROS by reduced form (CoQ10H2)	Cardiovascular diseases; Tumors	Listed Drug	[106]
Targeted mitochondrial antioxidant enzyme	Curcumin	Increasing the effect of superoxide dismutase, glutathione and catalase	Nervous system diseases; Tumors	Clinical Phase III	[107,108,109]
	AEOL 11207	Superoxide dismutase/catalase mimetics	Nervous system diseases	Leading compound	[110,111]
	EUK-418	Superoxide dismutase/catalase mimetics	Nervous system diseases	Preclinical	[110]
Others	Sulforaphane	Activating Nrf2	Nervous system diseases; Tumors; Cardiovascular diseases	Clinical Phase III	[112,113]
	Resveratrol	Activating Nrf2 and SIRT1; Increasing PGC-1α deacetylation	Nervous system diseases; cardiovascular diseases	Clinical Phase III	[114,115,116]

### 3.2. microRNA

MiRNAs are a class of short RNA molecules that can inhibit the translation of target messenger RNA (mRNA) or promote its degradation. MiRNAs are involved in almost all cell processes, and variations in their expression are linked to a variety of diseases. Mitochondrial-related miRNAs potentially play important roles in mitochondrial dysfunctions, and several studies have shown that miRNAs are involved in the regulation of mitochondrial processes under both normal and disease conditions [117]. MiRNAs can regulate the progression of AD by disrupting the mitochondrial membrane integrity to aggravate (miR-16–5p, miR-195, and miR-29b) [118,119,120] and inhibiting the activity of OXPHOS-related enzymes’ activity (miR-210, miR-338, and miR-34a) [81,121]. In addition, in AD transgenic mice and rat hippocampal neuronal cells, the silence of miR-204 can alleviate the symptoms of AD by reducing mitochondrial autophagy and ROS production in neurons, thereby [122,123]. In CVDs, Intravenous injection of microRNA-210 blocked the effect and recovered the increased myocardial IR injury and cardiac dysfunction by controlling mitochondrial bioenergy and ROS flux [124].

These results suggest that mitochondria-related miRNAs have potential application value in disease treatment, but the current research is mostly limited to cell level and animal models. The selection of miRNAs with real therapeutic application value still needs more research, especially the discovery of mitochondria-specific miRNAs with key regulatory functions.

### 3.3. Mitochondrial Transplantation

The function of mitochondria is regulated by multiple cytokines in the network, and most mitochondrial diseases are considered irreversible as they are caused by injury, such as mtDNA mutation. Antioxidants or drugs targeting only one target in the regulatory network only provide very limited protection in the treatment of mitochondrial-related diseases. Mitochondrial transplantation is a more useful strategy for the treatment of mitochondrial dysfunction to overcome the limitations of therapies using agents by replacement of non-functional mitochondria in damaged tissues or cells with functional ones.

A considerable number of studies reported the effectiveness of mitochondrial transplantation in various diseases in cell and animal models. Transfer of mitochondria obtained from bone marrow-derived mesenchymal stem cells (BM-MSCs) to renal proximal tubular epithelial cells (PTECs) of a diabetic neuropathy-induced animal model led to a decrease in ROS production and apoptotic cells [30]. Mitochondrial transplantation rescued liver function from APAP-induced hepatotoxicity in primary hepatocytes of the mouse with acetaminophen (APAP)-induced liver injury and resulted in amelioration of the liver by a decrease in both ROS production level and apoptotic cells in ischemia-reperfusion-injured rat liver [33,34]. In a heart disease model, the McCully group demonstrated that mitochondria translation from tissue unaffected by ischemia to the ischemic zone led to a significant enhancement in post-ischemic functional recovery and cellular viability and provided cardioprotection from ischemia-reperfusion injury [50,63]. In breast cancer cells or osteosarcoma cells, mitochondria translation from normal cells can also inhibit the proliferation of tumor cells, induce apoptosis, and reverse the carcinogenic characteristics of cancer cells [61,125]. In animal models, mitochondria isolated from the liver of normal mice can significantly inhibit the growth of melanoma and significantly prolong the survival period of tumor-bearing mice. Its anti-tumor mechanism is related to ATP depletion, apoptosis, and necrosis [126,127].

## 4. Conclusions

Mitochondria play a role in almost every aspect of cellular life. Currently, there is no consistent biochemical or molecular definition of a disease as a “mitochondrial disease”. However, mitochondrial dysfunction is a pathological process that occurs in a variety of diseases, including neurological diseases, CVDs, cancer, and metabolic diseases. Mitochondrial dysfunction plays different roles in the occurrence and development of different diseases, with cardiovascular and metabolic diseases being the most intensively studied. Due to limited research models and complex developmental regulatory networks, partial mechanisms of mitochondrial dysfunction in neurodegenerative diseases and cancer are still unknown. Mitochondrial-targeted drugs or therapeutic strategies have received widespread attention in mitochondrial-related diseases. At present, relevant researches focus on both antioxidants and mitochondrial translation. Although antioxidants have been used in clinical practice, they are still in the position of adjuvant therapy, and antioxidants are mostly found to have antioxidant effects in functional studies rather than from purposefully targeted screening. To a certain extent, mitochondrial transplantation is also limited in disease treatment because of its own technical difficulties and the complex role of mitochondrial disorders in disease. Therefore, the in-depth analysis of the regulatory mechanism of mitochondrial dysfunction, the discovery of key targets that can be used for screening, as well as the screening and drug design of inhibitors or agonists for specific regulatory proteins, and the discovery of mitochondria-related targeted transport system may be significant for the study of mitochondrial function and the treatment of mitochondrial related diseases.

## Figures and Tables

**Figure 1 biomedicines-11-00647-f001:**
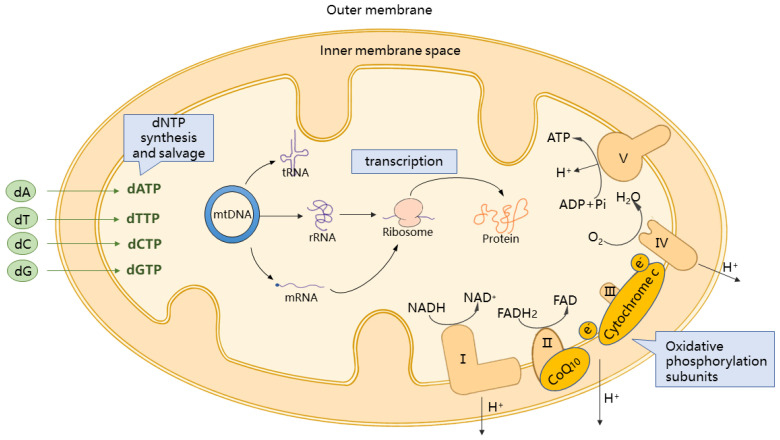
Illustration of mitochondrial oxidative phosphorylation system and other pathways. dA = deoxyadenosine. dC = deoxycytidine. dG = deoxyguanosine. dT = thymidine. dATP = deoxyadenosine triphosphate. dCTP = deoxycytidine triphosphate. dGTP = deoxyguanosine triphosphate. dNTP = deoxynucleoside triphosphate. dTTP = deoxythymidine triphosphate. mtDNA = mitochondrial DNA. Pi = phosphate.

## Data Availability

Not applicable.

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
