# Peer review of "A Mitochondrial Perspective on Noncommunicable Diseases"

_biomedicines, 2023, doi:10.3390/biomedicines11030647_

Round 1

Reviewer 1 Report

The manuscript of “A mitochondrial perspective on noncommunicable diseases” by Yujun Zhang and co-authors aims to review current knowledge about the structure, main functions of mitochondria, and role of the organelles in a number of human diseases, including neuromuscular diseases, cardiovascular diseases, metabolic diseases, neurodegenerative pathologies, and tumors. In a separate section, the authors discussed therapeutic strategies using mitochondria as therapeutic targets or targets of intervention.

The review is rather general, inconclusive and makes a weak contribution to the systematization of data on the underlying mechanisms of mitochondrial dysfunction and novel therapeutic approaches to treat mitochondrial diseases.

Comments:

1. The manuscript lacks originality and novelty. Sections 1,2,3 represent general traditional information about the structure and functions of mitochondria, but take up almost half of the manuscript. These sections may be deleted without prejudice to the scientific value and originality of the manuscript. The goal of the review should be specified.

2. The figures (Figure 1. Illustration of mitochondrial oxidative phosphorylation system and other pathways, and Figure 2. Mitochondrial dysfunction and human diseases) contain general information that can be found in textbooks on biochemistry, and have no value in the framework of modern science.

3. To summarize numerous experimental data and facilitate the perception of information by readers, it would be reasonable to add a table (with corresponding references) generalizing the underlying mechanisms of mitochondrial dysfunction in different tissues in human pathologies.

4. The section of “Therapeutic strategies using mitochondria as therapeutic targets or targets of intervention” could be improved by adding other novel methods (mitochondrial transplantation and others).

Author Response

  1. The manuscript lacks originality and novelty. Sections 1,2,3 represent general traditional information about the structure and functions of mitochondria, but take up almost half of the manuscript. These sections may be deleted without prejudice to the scientific value and originality of the manuscript. The goal of the review should be specified.

Thank you very much for your comment! We have deleted these content in the new manuscript, which is only described simply in the introduction.

  1. The figures (Figure 1. Illustration of mitochondrial oxidative phosphorylation system and other pathways, and Figure 2. Mitochondrial dysfunction and human diseases) contain general information that can be found in textbooks on biochemistry, and have no value in the framework of modern science.

Thank you very much for your comment! We have deleted Figure2 in the new manuscript and summarized the mitochondrial related diseases and antioxidants with two tables.

  1. To summarize numerous experimental data and facilitate the perception of information by readers, it would be reasonable to add a table (with corresponding references) generalizing the underlying mechanisms of mitochondrial dysfunction in different tissues in human pathologies.

Thanks, we summarized the mitochondrial related diseases and antioxidants with two new tables.

  1. The section of “Therapeutic strategies using mitochondria as therapeutic targets or targets of intervention” could be improved by adding other novel methods (mitochondrial transplantation and others).

Thank you very much for your comment! We have made major changes to the new manuscript, and we have also summarized the mitochondrial transport strategy again and added some data.

Reviewer 2 Report

General comments to the paper entitled: A mitochondrial perspective on noncommunicable diseases

Congratulation to the authors. The review excellently collects the most relevant papers, and the structure of the paper reflects the key issues relating to the connection between mitochondria dysfunction and different dieses.

Here are some minor changes I recommend

line 29: Figure 1.

line 159: misprint: Opal

line 199: (Figure 2.) should be in one line

line 235: Please give the number relating to Kann et al.

line 447: I suggest modifying the sentence “The presence of a large number of K+ channels in the IMM.”

line 453: extra space

line 486: misprint

Author Response

Here are some minor changes I recommend

line 29: Figure 1.

line 159: misprint: Opal

line 199: (Figure 2.) should be in one line

line 235: Please give the number relating to Kann et al.

line 447: I suggest modifying the sentence “The presence of a large number of K+ channels in the IMM.”

line 453: extra space

line 486: misprint

Thanks!We have revised the article and the relevant parts have been revised。

Round 2

Reviewer 1 Report

The manuscript has been substantially revised. I have no more comments.